# Potential Benefits of Probiotics and Prebiotics for Coronary Heart Disease and Stroke

**DOI:** 10.3390/nu13082878

**Published:** 2021-08-21

**Authors:** Haicui Wu, Jiachi Chiou

**Affiliations:** 1Department of Applied Biology and Chemical Technology, The Hong Kong Polytechnic University, Hung Hom, Kowloon, Hong Kong, China; wu.hc.wu@connect.polyu.hk; 2Research Institute for Future Food, The Hong Kong Polytechnic University, Hung Hom, Kowloon, Hong Kong, China

**Keywords:** CVD, probiotics, prebiotics, gut microbiota, immune homeostasis

## Abstract

Among cardiovascular diseases (CVDs), a major cause of morbidity and mortality worldwide, coronary heart disease and stroke are the most well-known and extensively studied. The onset and progression of CVD is associated with multiple risk factors, among which, gut microbiota has received much attention in the past two decades. Gut microbiota, the microbial community colonizing in the gut, plays a prominent role in human health. In particular, gut dysbiosis is directly related to many acute or chronic dysfunctions of the cardiovascular system (CVS) in the host. Earlier studies have demonstrated that the pathogenesis of CVD is strongly linked to intestinal microbiota imbalance and inflammatory responses. Probiotics and prebiotics conferring various health benefits on the host are emerging as promising therapeutic interventions for many diseases. These two types of food supplements have the potential to alleviate the risks of CVD through improving the levels of several cardiovascular markers, such as total and low-density lipoprotein (LDL) cholesterol, high sensitivity C-reactive protein (hs-CRP), and certain cytokines involved in the inflammatory response. In this review, we focus mainly on the preventive effects of probiotics and prebiotics on CVD via rebalancing the structural and functional changes in gut microbiota and maintaining immune homeostasis.

## 1. Introduction

In recent decades, CVD has been the principal contributor of early death and disability in low- and middle-income countries [1]. In most developed countries, over half of deaths in the middle-aged population and one-third of deaths in the elderly population are attributed to CVD [2]. CVD is a collective term, referring to a group of abnormalities in the CVS, and includes coronary heart disease (CHD, sometimes called ischemic heart disease or coronary artery disease), cerebrovascular disease, and peripheral vascular disease, which affect the blood supply to the heart, brain, and the peripheral regions of the body [2]. It is reported that half of the deaths from CVD are caused by CHD in the United States [3].

Unhealthy lifestyle, including an unbalanced diet, alcohol consumption, smoking, and physical inactivity, plays a crucial role in the etiology of CVD and is strongly related to CVD risk. The beneficial effects of probiotics and prebiotics on host health have been widely studied. The term probiotics originates from the Greek, meaning “for life”. The definition of probiotics established in 2014 by the joint Food and Agriculture Organization of the United Nations (FAO) and the World Health Organization (WHO) and accepted by the International Scientific Association for Probiotics and Prebiotics (ISAPP) is “live strains of strictly selected microorganisms which, when administered in adequate amounts, confer a health benefit on the host” [4]. The definition of “prebiotics” was first proposed by Gibson and Roberfroid in 1995 and further updated in 2004 as “non-digestible food components allowing the specificity of microbial changes in the intestinal tract, thereby exhibiting beneficial effects on host’s health” [5,6]. Currently, the most widely accepted definition of this term is reaffirmed by the International Scientific Association for Probiotics and Prebiotics (ISAPP) as a substrate that is selectively utilized by host microorganisms conferring a health benefit [4]. Probiotics and prebiotics are beneficial for human health and have long been recognized as potential dietary supplements to prevent the development of various bowel diseases, such as diarrhea and inflammatory bowel disease (IBD). In addition to the intestinal tract, where they pass through and may exert direct functions, accumulating evidence suggests that both probiotics and prebiotics could ameliorate metabolic disorders, including obesity, diabetes, and CVD [7,8,9]. Although functional foods have exhibited cardiovascular protective effects, the roles of probiotic and prebiotic supplements in the prevention and treatment of CVD are often based on brief reports and small-scale clinical studies, and the mechanisms have not been clearly elucidated. They have been indicated to protect against CVD by lowering cholesterol levels, attenuating oxidative stress, balancing functional and structural changes of gut microbiota, and improving immune responses [9,10,11].

Thanks to the advanced high-throughput techniques to sequence the gut microbiota, the role of gut microbiota in human health and well-being are being widely studied [12]. A growing body of evidence has demonstrated that the alteration of gut microbiota is linked to many diseases, including CVD [13]. The imbalance of intestinal microbiota has been observed in heart failure, thrombosis, atherogenesis, and arterial hypertension [14,15,16,17]. Probiotics and prebiotics playing positive roles in altering the microbial and metabolic composition of gut microbiota could be regarded as a potential therapeutic tactic for CVD.

The protective effects of probiotic and prebiotic therapies on CVD could also be explained by modulation of the host immune system. The immunological mechanisms underpinning probiotics and prebiotics involve the alterations of dendritic cells, epithelial cells, T regulatory cells, effector lymphocytes, natural killer T cells, and B cells [18]. Many chronic diseases are accompanied by low-grade inflammation, which is also the case for CVD. The plasma levels of proinflammatory factors IL-1, IL-6, and TNF-α, activated by innate and adaptive immune cells, are frequently found in CVDs [19].

This review aims to discuss the roles of probiotics and prebiotics as important dietary components in the development of major CVDs, namely CHD and stroke, via modulating the gut microbiota and immune system, based on available investigations.

## 2. CVD: Prevalence, Pathogenesis, and Risk Factors

Cardiovascular disease refers to any abnormalities involving the heart or blood vessels and includes CHD, stroke, high blood pressure, venous thrombosis, thromboembolic disease, cardiomyopathy, and arrhythmia [2]. The most well-known and extensively studied CVDs are CHD and stroke. As the leading cause of mortality at the global level, CVD contributes to 30% of all deaths across the world [1]. In addition, a report from World Health Organization (WHO) predicted that 23.6 million people will die due to CVDs, mainly from heart disease and stroke, by 2030. In the United States, approximately 43.9% of the population will develop CVD by 2030, as reported by the American Heart Association [20].

It has been shown that CVDs can be formed due to lesions on the coronary, cerebral, or peripheral arteries. The common pathophysiology in these diseases includes atherosclerosis, thrombosis, and clotting [21,22]. It is suggested that the initiation and progression of atherosclerosis are regulated by various immune responses [23]. Atherosclerosis is characterized by the presentation and progressive growth of atherosclerotic plaques (mainly lipid) in the walls of arteries. The lipid of those plaques is mainly cholesterol from LDL particles present in circulation. Lipoproteins enter the subendothelial space of arteries and activate endothelial cells. Meanwhile, monocytes in the vascular wall differentiate into macrophages, which engulf lipoproteins and become the so-called foam cells characteristic of atherosclerotic plaque [24]. Thus, atherosclerosis is a lipid-driven, chronic inflammatory disease, which is a principle predisposing factor for heart disease and stroke. The second common pathophysiology of CVD is clotting, which is initially produced in an inactive form, a precursor or zymogen, and subsequently depends upon a cascade of proteolytic reactions [2].

Since CVD represents a complex condition and involves various diseases, numerous risk factors have been reported to contribute to the occurrence of CVD. In general, the factors contributing to CVD can be categorized as non-modifiable risk factors and modifiable risk factors [25]. The non-modifiable risk factors are age, gender, and family history (genetics), while the modifiable factors include hypertension, smoking, diabetes mellitus, physical inactivity, obesity, unhealthy diet, cholesterol and lipids, depression and anxiety, and stress. Based on the risk factors for CVD, numerous therapeutic strategies—mainly medications and lifestyle modifications—have been suggested and applied [26,27,28]. The use of medications to prevent CVD-related diseases has drawbacks, including increased final burden in end organs, worsened compliance rates, and increased rates of side effects due to medication interactions [29]. These disadvantages have driven the development of new treatment options for CVD, highlighting the importance of adopting a healthy lifestyle, including consuming functional foods [30]. The bioactive compounds of functional foods are crucial for the prevention of CVD progression by altering the structure and function of gut microbiota and the associated immune responses [31,32]. Earlier literature has provided ample evidence that functional foods containing probiotics and prebiotics could prevent several cardio-metabolic disorders, including obesity, type I diabetes, and hypertension [7,8,33].

In this review, we combine the protective effect of probiotics and prebiotics on CHD and stroke via alteration of gut microbiota and modulation of host immune responses, providing implications for this complex disease with promising therapeutic strategies.

## 3. Gut Microbiota and CVD

Gut microbiota is a set of microorganisms colonizing in our gut and contains bacteria, archaea, viruses, and unicellular eukaryotes [34]. Bacteria, as part of this microbial community, have been most widely studied, to determine their roles in affecting human health. The number of bacteria cells in the gastrointestinal tract (GIT) is estimated to be approximately 3.8 × 10^13^, similar to that of total human cells in the body. Yet, the total mass of gut microbiota in a healthy individual is only 0.3% of the total body weight [35]. Intriguingly, there are 9 million unique genes present in the GIT, observed in the metagenomic studies, which is 450 times more than the whole genome in the human body [36,37].

A growing body of evidence indicates that gut microbiota is linked to the development of several cardio-metabolic diseases, such as diabetes mellitus, obesity, hypertension, and CVD [38]. Structural changes in the gut microbiota have been observed in various diseases. For instance, the ratio of *Firmicutes* and *Bacteroidetes* (F/B ratio), as well as the evenness, richness, and diversity of gut microbiota, are routinely reported to decrease in the aforementioned disorders [38,39,40,41]. Moreover, several unique bacterial species in the GIT have been reported to play crucial roles in human health. Recently, a promising candidate, *Akkermansia muciniphila,* was proposed to harbor the potential to be the so-called next-generation probiotic, despite not yet being included in the list of conventional probiotics. The relative abundance of *A. muciniphila* is positively associated with healthy status in comparison to patients with obesity and diabetes [42]. On the other hand, some cultivable microbes present in the gut have been long recognized to cause diseases, e.g., the diarrheagenic bacterial pathogens *E. coli*, *Salmonella*, *Campylobacter*, and *Shigella* species [43]. The microbiota in the gut, also viewed as “bioreactors”, could ferment food components and break them into functional metabolites or microbial products, such as short chain fatty acids (SCFAs) and secondary bile acids [44,45]. Thus, the structural and functional changes in gut microbiota play an important role in maintaining host health.

Intestinal microbiota and their metabolites are strongly associated with the progression of CVD [46]. Pathogens including *Shigella*, *Campylobacter*, *Yersinia*, *Streptococcus spp.*, as well as *Enterobacteriaceae* and *Candida*, were observed to be abundant in the stool samples of patients with CVD [47,48]. In addition, metagenomic and metabolomic analyses of fecal and plasma samples, respectively, in healthy populations and chronic heart failure (CHF) patients showed that the composition and metabolites of gut microbiota were significantly different. In this study, the abundance of *Faecalibacterium prausnitzii* was lower, while *Ruminococcus gnavus* was higher, in CHF than those in controls. The increase of butyrate and the decrease of trimethylamine N-oxide (TMAO) were also observed in CHF [49]. TMAO, one of the well-studied metabolites formed by gut microbiota, is positively correlated with early atherosclerosis [50]. This metabolite increases atherosclerotic plaque size, triggers prothrombotic platelet function, and promotes arterial thrombus growth. Another investigation suggested that the increased TMAO levels derived from gut microbiota through the metabolism of choline in female C57BL/6J *ApoE^-/-^* mice administered a choline diet promoted the formation of atherosclerosis [51]. In addition, the microbial cell wall component of Gram-negative bacteria, lipopolysaccharide (LPS), may compromise cardiovascular function and increase CVD risk. Intake of low-dose LPS in animal models resulted in vascular inflammation and atherosclerosis [52,53]. In addition, circulating endotoxemia (LPS) was most notable in those with the highest CVD burden. Resultant endotoxemia is associated with systemic inflammation, markers of malnutrition, cardiac injury, and reduced survival [54]. The primary bile acids are synthesized from cholesterol in the liver and further metabolized by the gut microbiota into secondary bile acids. A high level of secondary bile acids, produced solely by intestinal bacteria, in the enterohepatic circulation of some individuals may be involved in the progression of several diseases, including CVD [55]. Hence, the composition of commensal microbiota and their metabolites act as an emerging risk factor for CVD. 

## 4. Probiotics and Their Potential Role in CVD

### 4.1. Probiotics

Probiotics contain a set of beneficial microorganisms. Most of these belong to lactic acid bacteria, such as *Lactobacillus*, *Bifidobacterium*, *Lactococcus*, *Streptococcus*, and *Enterococcus*. The yeast genus *Saccharomyces* is also one of the well-known probiotics [56]. Currently, fermented foods, including yogurt, kefir, sauerkraut, and kimchi, serving as part of the human diet, are the main source of probiotic strains. There are several criteria for evaluating whether probiotics could be used in food in clinical research. These criteria are (1) proper identification, characterization, and maintenance of probiotic strains; (2) keeping the studied probiotics in live condition; and (3) ensuring they are alive at the site of action in the studies. In addition, the number of living microorganisms in foods containing probiotics at the time of human consumption should be above 10^6^ cells/mL or cells/g, according to the WHO, and the therapeutic dose in clinical study is 10^8^–10^9^ cells/mL or cells/g [57]. 

Unequivocal evidence demonstrates probiotics provide benefits for human health and play a crucial role in the prevention of various diseases [58]. The intake of probiotics could directly pass to the GIT and has been reported to be effective in the treatment of various types of diarrhea, especially traveler’s diarrhea, antibiotic-induced diarrhea, diarrheal diseases in young children caused by rotavirus, and inflammatory bowel disease (IBD), via counteracting the activity of pathogenic intestinal microbes and promoting the proper balance between pathogens and the commensal bacteria necessary for normal functions in the gut [59]. The potential application of probiotics also includes the prevention and treatment of cancer [60], food allergy [61], and atopic dermatitis [62]. The mechanisms of the beneficial effects of probiotics on human health or their preventive abilities against diseases are mainly due to competing with pathogenic microorganisms, antagonizing pathogens, altering gut microbiome, or modulating the immune response in the host [58]. While the ameliorative effects of traditional probiotics have been widely studied, recent studies have unraveled potential next-generation probiotics (NGPs) from gut microbiota [63]. For example, *Bacteroides fragilis* shows anticancer effects and attenuates inflammation [64], *Faecalibacterium prausnitzii* protects mice against intestinal diseases [65], and *Prevotella copri* and *Christensenella minuta* play a role in controlling insulin resistance [66].

### 4.2. Beneficial Effects of Probiotics on CVD 

CVD is a term representing a group of diseases and has a complex etiology for pathogenesis. Here, we discuss and summarize the role of probiotics in the well-known CVDs, namely CHD and stroke. We searched from the database “PubMed” using keywords “probiotics and coronary heart disease or coronary artery disease” from 2000 to 2021. A total of 83 relevant articles were present, but only 12 studies were selected according to title and abstract. Those studies showed that probiotic strains exhibited beneficial effects on CHD (Table 1). Three of these investigations, reported by Raygan et al., indicate that the supplementation of probiotics and co-supplementation with probiotics and vitamin D or selenium for diabetic patients with CHD could significantly improve the biomarkers of mental health and metabolic profiles, such as hs-CRP, nitric oxide (NO), LDL or total cholesterol, and the parameters involved in inflammation and oxidative stress [67,68,69]. Another two articles report that supplementation with synbiotics, containing probiotic strains and prebiotic “inulin”, in patients with CHD displayed improvement in the levels of several biomarkers, including serum hs-CRP, cholesterol, plasma NO, and malondialdehyde (MDA) [70,71]. The other three reported articles are linked to coronary artery disease (CAD). *L. plantarum* 299v (Lp299v) supplementation ameliorated vascular endothelial function and attenuated system inflammation in men with CAD [72]. The intake of *Lactobacillus rhamnosus* GG (LGG) also exhibited beneficial effects in reducing metabolic endotoxemia and mega inflammation in participants with CAD [73]. Another study conducted by the same group found that co-supplementation of probiotics and inulin in CAD subjects for eight weeks had beneficial effects on depression, anxiety, and inflammatory biomarkers [74]. On the other hand, the beneficial roles of probiotics on stroke have been researched, and four articles were found (Table 1). However, all the benefits of probiotic strains on stroke were observed in rodent models. In mice, 2 weeks of treatments with *Clostridium butyricum* and a probiotic mixture including *Bifidobacterium breve, Lactobacillus casei, L. bulgaricus (L. delbrueckii subsp. bulgaricus)*, and *L. acidophilus* improved ischemia via the mechanisms involved in anti-oxidation and anti-inflammation and ameliorated neurological deficits [75,76]. The intake of probiotic *Bacillus licheniformis* CMCC 63516 showed preventive effects on heat stroke in rats by sustaining intestinal barrier function, such as increasing tight junctions and decreasing intestinal injury and modulating gut microbiota, e.g., increasing the ratio of *Lactobacillus* and *Lactococcus* in the gut microbiota [77]. Moreover, the protective effect of Lactobacillus *amylovorus* DSM 16698T (ILA) on cerebral ischemia reperfusion injury in rats was observed through attenuating cerebral infarction volume and neural cell apoptosis, decreasing the levels of MDA and TLR-4, and increasing superoxide dismutase (SOD) activity [78]. The protective roles of probiotics as potential dietary supplementation against other CVDs have also been investigated, though those studies are not presented in this review.

### 4.3. Underlying Mechanisms of Probiotics on CVD

Sánchez et al. proposed four different mechanisms contributing to the beneficial effects of probiotics on human health: (1) amelioration of the epithelial barrier function; (2) competing against pathogens with nutrients and adhesion sites; (3) effect on other tissues by the immune system and neurotransmitter production; and (4) immunomodulation [79]. The underlying mechanisms of probiotics on CVD are relatively complicated and are yet to be elucidated. Here, we discuss the preventive effects of probiotics on CVD via restoration of gut microbiota dysbiosis and anti-inflammatory responses, though the mechanisms of the beneficial effects of probiotics on CVD likely include reducing oxidative stress, lowering hypercholesterolemia, and lowering high blood pressure [9].

(1)From the view of gut microbiota

The changes in gut microbiota, which is associated with the mediation of cholesterol metabolism, uric acid metabolism, oxidative stress, and inflammatory reactions through various metabolites, could be involved in the development of atherosclerosis, the major risk factor for CHD and stroke [80,81,82,83]. A study conducted by Karlsson et al. using a whole-genome sequencing approach suggested that there is a possible link between gut microbiota changes and atherosclerotic heart disease [84]. The abundance of *Collinsella* was higher, while that of *Rothia* and *Eubacterium* spp. was lower, in these patients than in their counterparts in healthy controls. Moreover, the F/B ratio significantly decreased in people with CHD conditions. Later, the proportion of *Lactobacilli* increased, while *Bacteroidetes* including *Bifidobacterium* and *Prevotella* decreased notably, in patients with CHD compared to healthy volunteers, as reported by Emoto et al. in 2016 and 2017 [85,86]. In several animal-based studies, stroke-induced shifts in the structure of gut microbiota included an increased abundance of *Clostridial* and *A. muciniphila* [87]. In addition, the alteration of intestinal microbiota, such as reduced species diversity and overgrowth of *Bacteroidetes*, was observed in another animal stroke model, the mice middle cerebral artery occlusion model [88]. In patients with ischemic stroke, a reduction in the diversity and increased abundance of *Bacteroidetes* phylum in the gut microbiota was also found [89].

Current therapies targeting the dyslipidemia associated with atherosclerosis have various side effects. In comparison to pharmaceutical agents, nutraceuticals generated from the food after fermentation by probiotic bacteria have evoked greater interest in atherosclerosis prevention [90]. Supplementation of probiotic *L. mucosae* A1 exhibited effectiveness in the treatment of hyperlipidemia and atherosclerosis and improved gut microbiota dysbiosis, including increased richness and diversity in ApoE ^-/-^ mice on a Western diet [91]. The anti-atherosclerotic effects of *L. plantarum* ATCC 14917 were also observed in ApoE ^-/-^ mice via modulation of proinflammatory cytokines TNFα and IL1-β, oxidative stress, and gut microbiota [92]. Administration of this probiotic strain increased the proportion of *Bacteroidetes*, whereas it attenuated the abundance of *Firmicutes*, *Verrucomicrobia,* and *Proteobacteria*. At the family level, *Bacteroidaceae* was significantly increased, whereas other major families were decreased by this treatment. Moreover, the exopolysaccharide-producing probiotic *Lactobacilli mucosae* DPC 6426 ameliorated atherosclerosis by reducing serum cholesterol and altering the relative abundance of enteric microbiota, including greater prevalence of *Porphyromonadaceae* and *Prevotellaceae* and lower abundance of *Clostridiaceae*, *Peptococcaceae*, and *Staphylococcaceae* in the DPC 6426 group compared with the placebo group [93]. Another study indicated that the ApoE ^-/-^ mice supplemented with the probiotic mixture VSL#3 (a mixture of *S. thermophilus, B. breve, B. bacterium longum, B. infantis, L. acidophilus, L. plantarum, L. paracasei*, and *L. delbrueckii subsp. bulgarius*) exhibited different intestinal microbiota composition from that of the control groups [94].

Apart from the structural changes in enteric microbiota, several gut-derived metabolites, especially TMAO, are strongly correlated with CHD risk [95,96]. Trimethylamine (TMA) is formed from choline, carnitine, and phosphatidylcholine in foods via the intestinal microbial enzyme complex and the carnitine Rieske-type oxygenase/reductase system in the GIT [97,98]. After this compound is formed, it enters the portal vein circulation and is metabolized by the liver enzyme, flavin-containing monooxygenase, to further become TMAO. Probiotics play a potential role in preventing atherosclerosis through the amelioration of TMAO [99]. The development of TMAO-induced atherosclerosis was considerably blocked by *L. plantarum* ZDY04 in ApoE^-/-^ mice treated with 1.3% choline as compared with the control group [100]. Moreover, microbiota metabolic changes could be observed with the introduction of probiotics. Administration of fermented milk containing *B. animalis subsp. lactis* lowered the proportion of several harmful bacteria and modulated the production of SCFAs in the colon [101]. CVDs, such as dyslipidemia and diabetes, could be caused by disorders in the metabolism of bile acid. *Lactobacilli* play an important role in BA biotransformation by promoting the activity of microbial bile salt hydrolase (BSH), regenerating primary free bile acids, and facilitating the microbial formation of secondary BAs as well as a range of intermediates [55,102]. Therapeutic manipulation of microbiota using different antimicrobial strategies may be a useful approach for the management of cardiovascular-related diseases. *Lactobacilli* or other lactic acid bacteria are excellent antimicrobial producers and might improve CVD by promoting the growth of beneficial bacteria and inhibiting the production of LPS by Gram-negative bacteria in the GIT.

Although many studies have suggested that probiotics showed preventive effects on CVD, particularly on the development of atherosclerosis, through modulation of gut microbiota, the underlying mechanisms are not clearly elucidated. Here we propose that treatments with different probiotic strains on CVD patients or animals may rebalance bacteria composition and their metabolites in the gut by directly interacting with other beneficial bacteria and competing against harmful microorganisms, or indirectly helping/inhibiting the production of gut-derived metabolites, thereby affecting the downstream inflammatory responses or gene expressions in end-organs of the host (Figure 1).

(2)From the view of inflammatory response

Many chronic diseases, including CVD, are involved in low-grade inflammation. Frequently, the plasma levels of the main proinflammatory mediators, TNF-α, IL-1, and IL-6, are higher in cardiovascular disorders as compared with healthy conditions. The inflammatory biomarkers of CHD include IL-6, C-reactive protein (CRP), complement system, cluster of differentiation 40 (CD40), and myeloperoxidase (MPO) [103]. The abovementioned research regarding the role of probiotics in CHD revealed that administration of pure probiotic strains or the co-supplementation of probiotics with other products attenuated levels of hs-CPR and other inflammatory factors [67,68,69,70,71]. In addition, the anti-inflammatory effects of probiotics on stroke by decreasing the expression of tight junction proteins and inflammatory cytokines such as TNF-α were observed [76,77,78]. Although these studies demonstrated that the introduction of probiotics is associated with anti-inflammatory effects on CHD and stroke, the mechanism of this action is yet to be investigated.

The consumption of probiotics exhibited beneficial effects on the host through stimulating both the innate and adaptive immune system [104]. The ability of adhesion to intestinal epithelial cells (IECs) is one of the criteria to evaluate probiotics. *L. casei* CRL 431 and *L. paracasei* CNCM I-1518 adhere to IECs through the Toll-like receptors (TLRs) and modulate immune response, such as increasing the production of IL-6 and macrophage chemoattractant protein 1 (MCP1) [105]. The intestinal barrier could also be strengthened by probiotics via multiplying the number of Goblet cells, which results in the reinforcement of the mucus layer [106]. Some probiotic strains have been reported to increase the expression of mucin-related proteins, such as MUC2, MUC3, and MUC5AC, in HT29 cells [107]. Oral administration of probiotics also increases the number of Paneth cells and reinforces the intestinal barrier integrity through increased gene expression of tight junction signaling [108,109,110]. These modulations were involved in anti-inflammatory responses of probiotics. The intake of probiotic strains, such as *L. plantarum* 299v and *L. rhamnosus* GG, could improve CVD via alleviating inflammatory responses, which include decreased IL-8, IL-12, IL-1β, and TNF-α in patients [72,73,74]. In addition, it has been reported that the induction of probiotics *C. butyricum*, *B. breve*, *L. casei*, *L. bulgaricus*, *L. acidophilus*, and *Bacillus licheniformis* showed preventive effects on stoke along with reduced levels of inflammatory cytokines in both mice and rats [75,76,77,78]. In addition, probiotics could alter specific commensal bacteria of the gut microbiota, such as *Prevotella* and *Oscillibacter*, which have been shown to produce anti-inflammatory metabolites and thus decrease Th17 polarization and promote the differentiation of anti-inflammatory T_reg_/Type 1 regulatory T (Tr1) cells in the gut [111].

Approximate 50% of stroke patients are associated with gastrointestinal complications, which include gut dysmotility, leaky gut, dysbiotic gut microbiotia, and even gut-origin sepsis; these patients have poor stroke outcomes [112]. The underlying mechanisms remain understudied. One possible explanation is the bidirectional communication between the gut and the nervous system: The so-called gut–brain axis. Sympathetic nervous system (SNS) activation is associated with inflammation-induced vascular endothelial dysfunction and cardiometabolic disease [113]. Since some gut bacteria, such as *Bifidobacterium* and *Lactobacillus* genera, could produce neurotransmitters, including γ-aminobutyric acid (GABA), serotonin, dopamine, and norepinephrine, supplementation of probiotics could cause compositional changes of gut microbiota so as to alter the types and concentrations of neurotransmitters, in turn influencing the SNS and subsequent cardiometabolic activities [114,115].

The potential mechanisms of probiotics’ beneficial effects on CVDs from the approach of immune responses are shown in Figure 1. Probiotic bacteria interact with IECs or immune cells through TLRs to (1) stimulate mucin production, (2) increase the expression of tight junction proteins, and (3) increase the number of Goblet and Paneth cells, which collectively enhance the intestinal barrier and result in activation of innate and adapted immune response, including production of various cytokines, chemokines, and anti-inflammatory metabolites (Figure 1).

## 5. Prebiotics and Their Potential Role in CVD

### 5.1. Prebiotics

There are several selection criteria for prebiotics, which are (1) resistance to digestion in the upper sections of the alimentary tract, (2) selective fermentation by potentially beneficial microbiota in the colon, (3) beneficial effect on host health, (4) selective stimulation of growth of probiotics, and (5) stability in various food/feed processing conditions [116].

Based on the definition and criteria proposed, a set of potential prebiotics has been recognized over the past decades. Those prebiotics, according to the number of monomers bound together, are sorted into disaccharides, oligosaccharides, and polysaccharides [117]. Research has demonstrated that oligosaccharides, especially fructans (fructooligosaccharides (FOS) and inulin) and galactans (galactooligosaccharides (GOS)), are the well-known prebiotics, as evidenced by in vitro and in vivo studies. Dietary fiber is also non-fermentable by human digestive enzymes and is sometimes used interchangeably with prebiotics. Dietary fiber is fermentable by the majority of colonic bacteria, which are not well-defined or fermented at all, whereas prebiotics could be used by the strictly defined microorganisms in the colon [118]. Numerous health benefits have been found as a result of the supplementation of prebiotics in the diet. Some investigations have suggested that people consuming high amounts of fiber or other types of prebiotics exhibited considerably lower risk of diabetes, reduced body weight, and decreased colorectal cancer prevalence [119,120,121].

### 5.2. Beneficial Effects of Prebiotics on CVD

The consumption of prebiotics protects the host against CVD. Using keywords (“prebiotics” or “inulin” or “fructans” or “galactans”) and (“cardiovascular diseases” or “coronary heart disease” or “stroke”) in PubMed from 2000–2021, we found 167 relevant results, but only 11 articles discussing the roles of prebiotics on CVD after screening title and abstract (Table 2). Among these, most studied the beneficial effects of inulin or the prebiotics containing inulin on various CVDs, including CHD or diabetes associated with CHD, coronary artery disease, chronic kidney disease, atherosclerosis, and hypercholesterolemia, in both human patients and animal models [70,71,74,122,123,124]. Supplementation with inulin (or co-supplementation with inulin and other components) reduced the levels of cholesterol, including total and LDL cholesterol, CRP, and several inflammatory cytokines, and improved anti-oxidative parameters and gut microbiota dysbiosis. Nevertheless, in contrast to previous results, which showed that inulin supplementation is effective at lowering inflammation and plasma lipid levels, one article indicated that inulin aggravated the accelerated atherosclerosis development driven by increased plasma cholesterol in hypercholesterolemic APOE*3-Leiden mice [125]. In addition to inulin, the beneficial roles of other prebiotics or prebiotic complexes in CVD were also reported. The administration of a prebiotic complex based on fermented wheat bran was observed to correct intestinal dysbiosis and endotoxemia in female rats with modeled heart failure [126]. The beneficial effects of dietary supplementation with soluble fiber (Minolest) on the lipid profile in subjects with mild hypercholesterolemia and a low risk of coronary artery disease were determined [127]. Another study found that larch arabinogalactan, an active component of pectin, attenuated myocardial injury by inhibiting apoptotic cascades in a rat model of ischemia-reperfusion [128]. In addition, chitosan oligosaccharides (COS) exhibited protective effects in CHD by ameliorating antioxidant capacities and lipid profiles through promoting the growth of probiotic species in intestinal flora [129]. Altogether, prebiotics could improve symptoms of CVD via several mechanisms involved in inflammation, antioxidant capacity, and rebalancing of the dysbiotic gut microbiota. However, the adverse effects of prebiotics on CVD reported necessitate caution in the application of inulin in humans.

### 5.3. Underlying Mechanisms of Prebiotics on CVD

The mechanisms of action of prebiotics on host health have been proposed. First, prebiotics present in natural products or added into food could be fermented by gut bacteria and thus affect the structure of gut bacteria composition and the metabolic activity [130]. Second, prebiotics are able to change the environment in the gut, thereby reducing the pH and inhibiting the growth of pathogens in the colon [131]. The third mechanism is the beneficial effect on immunological function, though the evidence of this mechanism remains unclear. Several plausible explanations for the potential mechanisms of prebiotic beneficial effects on CVD have been proposed, which will be discussed and summarized in terms of two main aspects—modulation of gut microbiota and inflammatory responses [132].

(1)From the view of gut microbiota

Prebiotics improve human health by maintaining a balanced gut microbiota and restoring its homeostasis [133,134,135]. They are fermented mainly by saccharolytic bacteria and affect the composition of gut microbiota, especially favoring the growth of beneficial bacteria such as *Lactobacillus* and *Bifidobacterium*. These bacteria can also block the proliferation of harmful bacteria [133,134]. The beneficial effects of a prebiotic complex based on fermented wheat bran and prebiotic chitosan oligosaccharides on heart failure and CHD are partially attributed to the rebalancing of gut microbiota dysbiosis and promoting the growth of different probiotic species [126,129]. The metabolites of intestinal microbiota could be altered by prebiotics through the fermentation process. SCFAs including acetate, propionate, and butyrate are the main end-products of prebiotic fermentation by the gut bacteria. These metabolites show health-promoting functions, including lowering glycemic levels and body weight and improving intestinal membrane integrity [136]. SCFAs bind to deorphanized G protein coupled receptors, such as free fatty acid receptors (FFAR) 2 and 3 and olfactory receptor 78 (Olfr78), which exhibit higher affinity for acetate and propionate, and GPR 109A, which has a higher affinity for butyrate [137,138,139]. Another function of SCFAs is to inhibit histone deacetylase. Histone deacetylase inhibitors (HDACs) can regulate chromatin structure to activate transcription factor and downstream gene expression [136]. SCFAs have been shown to be inversely associated with some risk factors for CVD. Acetate and propionate have been demonstrated to regulate blood pressure through Olfr78 and FFAR3, as seen in knockout animal models [139,140]. Butyrate attenuated blood pressure in angiotensin II-induced hypertensive rats, mainly via reducing the expression of renal protein receptors and renin [141]. Hyperglycemia in obesity and insulin resistance triggers increased gut permeability, which contributes to an inflammatory cascade [142]. SCFAs are crucial for maintaining a healthy gut, particularly in modulating epithelial integrity via tight junction proteins. For example, butyrate regulates proteins of the tight junction complex via acting on nucleotide-binding oligomerization domain-like receptors (NLRs), which subsequently modulate inflammation [143]. Additionally, SCFAs play a role in appetite regulation and energy intake to protect against obesity [144]. Glucose homeostasis can also be modulated by SCFA through improved insulin sensitivity via ameliorated gut barrier function and increased anti-inflammatory and antioxidant abilities [145].

Altogether, prebiotics, especially fibers or their end-product SCFAs in the gut, exhibit beneficial effects on CVD through several mechanisms involved in alteration of the gut environment, histone deacetylation, improvement of gut epithelial permeability contributing to reduced total and LDL cholesterol and hs-CRP, and lowering the incidence of CVD risk factors such as hypertension, obesity, and diabetes (Figure 2).

(2)From the view of inflammatory response

On the other hand, prebiotics play an important role in modulation of the immune response and host defenses [133,134]. Different types of prebiotics showed various immunological functions in innate and adaptive immune systems. Dendritic cells are the cell types most reported to be influenced by prebiotics. Three major forms of SCFAs, as the main metabolites from the prebiotics in the colon, exhibit different immunomodulatory effects. Expression of Foxp3 by dendritic cells can be induced by propionate, but not acetate, possibly due to the lack of HDAC activity [146]. Among SCFAs, butyrate is more effective than propionate and acetate in inducing immunomodulation. Moreover, it reduces the secretion of dendritic cell IL-12 and IL-6 cytokine and promotes T_reg_ cells by dendritic cells [146]. Another study also confirmed that butyrate induced T_reg_ cells and IL-10-secreting T cells through binding to GPR109a on colonic dendritic cells and macrophages [138]. In the adaptive immune system, SCFAs also show beneficial effects on lymphocytes. The introduction of a mixture of 17 Clostridia strains ameliorated colitis and allergic diarrhea in mice due to production of SCFAs that bound to GPCR 43 and subsequently attenuated those diseases through a T_reg_/TGF-β-dependent mechanism [147,148]. In addition, gut microbiota may influence brain activity and host physiological status through SCFAs, since these microbially derived products are involved in the signaling to the brain via nerve activation. Butyrate and propionate can regulate neurotransmission by increasing expression of tyrosine hydroxylase, which is important for dopamine and noradrenaline synthesis [149]. Evidence also suggests that propionate can lower levels of the indoleamine serotonin GABA as well as dopamine [150]. Thus, SCFAs could regulate the immune system through the gut–brain axis.

Only a few articles have reported on the immunomodulatory function of prebiotics on CVD. Specific prebiotics, GOS, FOS, and pectin-derived acidic OS, exhibited an immunomodulatory effect during the early phase of a murine immune response in C57BL/6 mice [151]. Nutritional supplementation with prebiotics in elderly persons reduced the levels of pro-inflammatory factors, leading to an overall decrease in gut inflammation [152,153]. One abovementioned study also highlighted the potential role of prebiotics in CVD via improving proinflammatory cytokines [74]. Another study indicated that inulin treatment improved atherosclerosis and the number of macrophages in male APOE*3-Leiden mice [125]. The association between inflammation and cardiovascular risks has been supported by much research. Thus, prebiotics have a positive effect in preventing the development of CVD, partially through their anti-inflammatory action.

## 6. Beneficial Effects of Postbiotics on CVD

Though the precise definition of postbiotics is under discussion, according to Tsilingiri et al., postbiotics include any substance released by or produced through the metabolic activity of the microorganism, which exerts beneficial effects on the host, directly or indirectly [154]. Postbiotics cannot be considered as synbiotics based on the current literature. The concept of postbiotics strengthens the beneficial effects of the microbiota through the secretion of various bioactive metabolites. Postbiotics, although not containing live bacteria, show benefits on host health via mechanisms similar to probiotics. In general, the currently available classes of postbiotic substances include (1) cell-free supernatants that have biologically active metabolites secreted by bacteria and yeast into the surrounding liquid; (2) exopolysaccharides produced by biopolymers, released from microorganisms; (3) antioxidant enzymes generated from microorganisms to protect against the harmful effects of ROS; (4) cell wall fragments, including bacterial lipoteichoic acid (LTA); (5) SCFAs, which are the products of plant polysaccharide breakdown by intestinal microbiota; (6) bacterial lysates obtained by the chemical or mechanical degradation of Gram-positive and Gram-negative bacteria; (7) metabolites produced by gut microbiota [155]. These postbiotics display many beneficial effects on the human body, such as immunomodulatory effects, antitumor effects, infection prevention, and accelerated wound healing. It is of interest that postbiotics have antiatherosclerotic effects. For example, Kefiran, an exopolysaccharide produced by *Lactobacillus kefiranofaciens,* prevented the onset and development of atherosclerosis in hypercholesterolemic rabbits, resulting in the reduction of inflammation and lipid concentration as well as the prevention of cholesterol accumulation in macrophages [156]. In addition, fragmented lactic acid bacterial cells from *Lactobacillus amylovorus* CP1563 might be beneficial for the prevention and treatment of dyslipidemia by attenuating the levels of LDL cholesterol and triglycerides while increasing the level of beneficial HDL cholesterol in an obese mouse model [157]. Although the potential mechanisms of postbiotic action on CVD have not been elucidated, it is intriguing to investigate the beneficial roles of postbiotics in this disease.

## 7. Role of Probiotics and Prebiotics in Other CVD: Hypertension

Hypertension is one of the most important risk factors of CVD. The therapeutic effects of probiotics, including probiotic strains and products fermented by live probiotics, on hypertension in both animal models and humans have been reported [33,158,159]. Orally administered pure strains, such as those belonging to *L. plantarum*, *L. fermentum*, *L. coryniformis*, *L. gasseri*, *L. casei*, *B. longum*, *B. breve*, and *B. infantis*, at optimal doses could alleviate high blood pressure in hypertensive patients and animals [160,161,162,163]. The antihypertensive effects of probiotics, especially lactic acid bacteria, contribute to an array of proteolytic cassettes and peptide transporters [164]. The most studied fermented milk products for their hypotensive effect are yogurt, cheese, and kefir, which contain live probiotic strains. For example, the consumption of fermented milk reduced systolic blood pressure, probably through milk-derived tripeptides isoleucine-proline-proline (Ile-Pro-Pro) and valine-proline-proline (Val-Pro-Pro), via inhibiting ACE in both hypertensive patients and spontaneously hypertensive rats (SHRs) [165,166]. Their antihypertensive mechanisms involve production of fermented food–derived bioactive peptides or polyphenols to further regulate RAAS, rebalance gut microbiota, and reduce inflammatory responses.

In addition, a growing body of evidence indicates that prebiotics, especially fiber and its end-product fermented by gut microbiota, were able to ameliorate hypertension. A diet high in fiber changed the gut microbiota and prevented the development of hypertension and heart failure in mice. The benefits of fiber might be due to the generation of one type of metabolite produced by gut microbiota: acetate. Acetate intake in this study effects several molecular changes associated with improved cardiovascular health and function [167]. The other two main SCFAs, propionate and butyrate, were also reported to improve hypertension and its associated cardiovascular damage [141,168,169,170]. The antihypertensive effects of these SCFAs might be through suppressing renal (pro) renin and the intrarenal renin-angiotensin system or inhibiting the COX2/PGE2 pathway via a HDAC5/HDAC6-dependent mechanism.

## 8. Future Perspectives

Abundant studies have been conducted to investigate the roles of probiotics and prebiotics in different diseases, including CVD. Despite being generally recognized as safe (GRAS) dietary supplements, with approval from the Food and Drug Administration (FDA) not required, the effects of probiotics and prebiotics on bloating, flatulence, and high osmotic pressure need to be further investigated [171]. In addition, specific dosages of probiotics and prebiotics should be considered to avoid potential adverse reactions. Although it is evident that consumption of probiotics and prebiotics is beneficial for CVD and promising as therapeutic strategies to manage conventional cardiovascular disease, contradictory findings are also reported in the literature, which are not presented in this review as we focused on CHD and stroke. There are still quite a few limitations in the use of probiotics, such as safety issues in vulnerable populations, limited studies evaluating the viability of live bacteria once in the intestine and the differences between viable or non-viable microorganisms, and lack of convincing evidence from clinical trials for certain indications. The recommendations from recent surveys on probiotic prescribing practices among health care providers and a review of current guidelines and published large clinical trials appear to be non-specific and inconsistent. Issues with the use of live microorganisms, especially the safety issues due to translocation of bacteria from the gut to the systemic circulation, lead to an increased interest in the use of postbiotics, which could be a safe and feasible strategy for the management of different diseases. However, the use of postbiotics in CVD is a relatively new area, and limited studies have been reported. In future, the number of investigations focusing on the health benefits of prebiotics and the possible mechanisms of action involved might be increased.

The introduction of probiotics and prebiotics could alleviate CVD via improving gut microbiota dysbiosis. Nevertheless, the interaction between supplemented probiotics and indigenous gut bacteria still needs to be clarified. Gut microbiome investigations are still complicated and far from standardized. In order to clearly elucidate the role of gut microbiota on CVD, sample sizes need to be adequate and proper control groups are required. Moreover, the positive effect of probiotics and prebiotics on CVD through anti-inflammatory modulation has been demonstrated, yet the underlying mechanisms remain largely unclear, and more evidence is needed.

## 9. Conclusions

In this review, we summarized the published results and proposed novel mechanisms that could contribute to the development of CVD, such as the metabolites produced from probiotics or prebiotics involved in immunomodulation via the gut–brain axis. Due to the limitations of probiotics and prebiotics, we also included a discussion of postbiotics, which is an intriguing field and a promising therapeutic strategy for the prevention of CVD in the future. CVD is undoubtedly linked to an increased risk of morbidity and mortality across the world. The intake of probiotics and prebiotics plays an important role in preventing and delaying the development of this disease. The levels of crucial CVD markers, namely LDL cholesterol and CRP, are significantly improved by the introduction of various probiotic strains and prebiotics. The underlying mechanisms of their protective effect in CVD, especially CHD and stroke, have been proposed. Furthermore, numerous studies suggest that probiotics and prebiotics protect against CVD by altering gut microbiota and intervening in inflammatory responses. These two categories of supplements could restore the normal gut microbiota, promote the growth of beneficial bacteria, and inhibit the proliferation of pathogens. Metabolites produced by gut microbiota, such as TMAO and SCFAs, could be modulated by pro- and prebiotics in patients having CVD. SCFAs, as the main microbial metabolites from prebiotics, are regarded as one of the major mediators in the communication between gut microbiota and the immune system, exhibiting immunomodulatory potential. To conclude, this review summarized the major findings from the literature to illustrate the potential underlying mechanisms of the beneficial effects of pro- and prebiotics, which could be considered as a promising intervention strategy to prevent or improve CVD via targeting the gut microbiota and maintaining immune homeostasis in the field (Figure 3).

## Figures and Tables

**Figure 1 nutrients-13-02878-f001:**
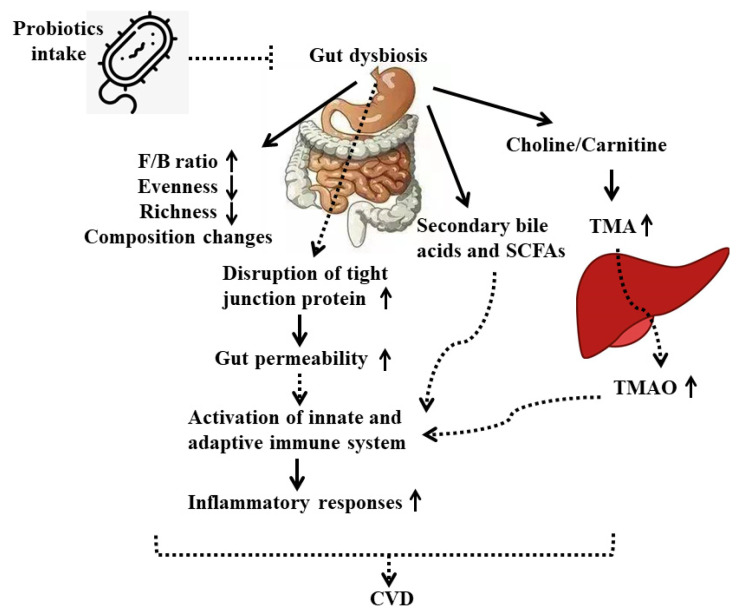
Mechanisms of the beneficial role of probiotics on CVD. Solid line represents direct outcome. Dashed line represents indirect outcome. F/B ratio, *Firmicute**s* to *Bactero**idetes* ratio; TMA, trimethylamine; TMAO, trimethylamine N-oxide.

**Figure 2 nutrients-13-02878-f002:**
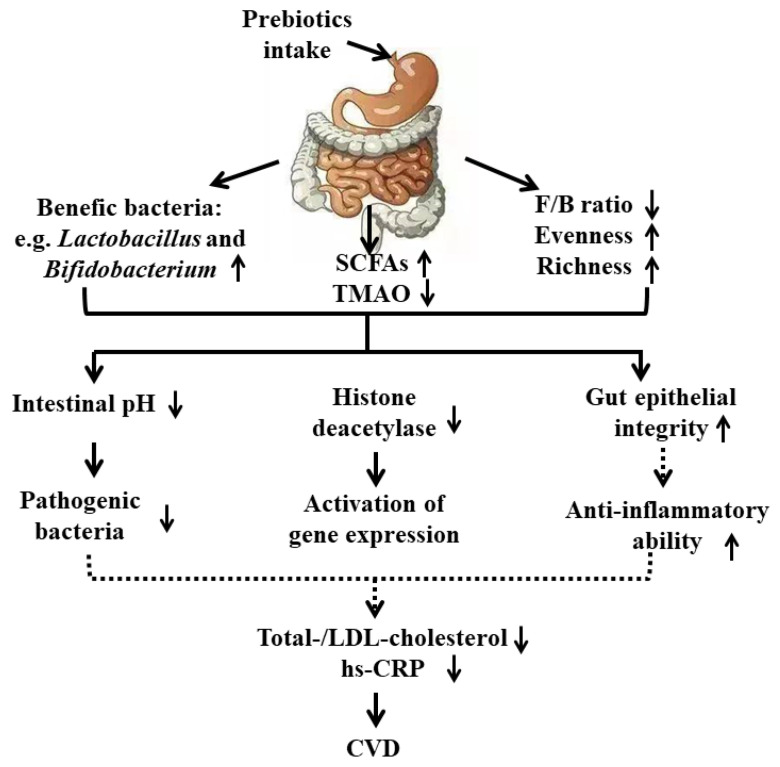
Mechanisms of the beneficial role of prebiotics on CVD. Solid line represents direct outcome. Dashed line represents indirect outcome. SCFAs, short chain fatty acids; LDL-cholesterol, low density lipid-cholesterol; hs-CRP, high sensitivity C-reactive protein.

**Figure 3 nutrients-13-02878-f003:**
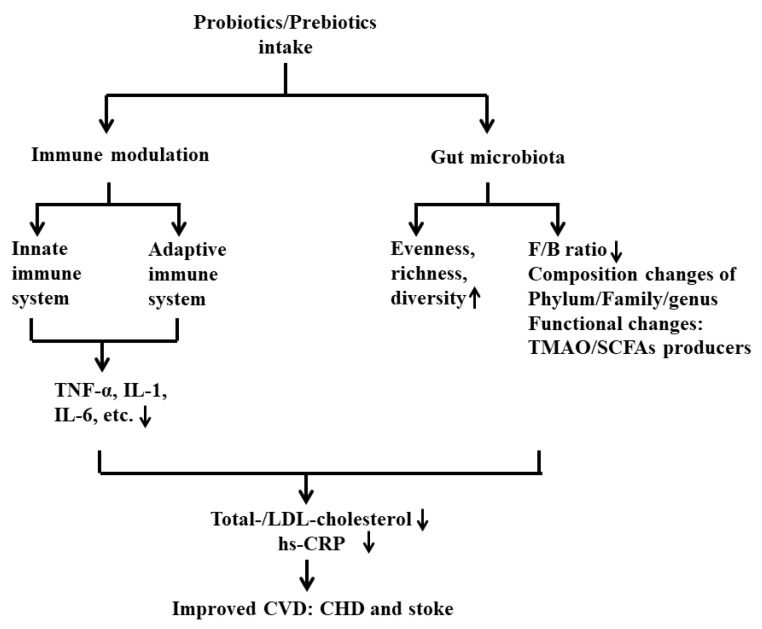
Summary of the mechanisms of the beneficial role of probiotics and prebiotics in CVD via gut microbiota and immune responses.

**Table 1 nutrients-13-02878-t001:** Beneficial effects of probiotics on CHD and stroke.

Disease	Product	Probiotic Strains	Subject	Dose	Duration	Outcomes	Reference
Type 2 diabetic patients with CHD	Probiotic	*B. bifidum*, *L. casei*, *L. acidophilus*	Human	2 × 10^9^ CFU/day	12 weeks	Fasting plasma glucose, insulin, insulin resistane and total/HDL cholesterol ratio ↓ Insulin sensitivity and HDL cholesterol levels ↑, hs-CRP ↓ Antioxidant capacity and total glutathione levels ↑	[67]
Diabetic people with CHD	Vitamin D and probiotics	*L. zisttakhmir*	Human	8 × 10^9^ CFU/g	12 weeks	Serum insulin levels ↓ Serum 25-OH-vitamin D ↑ Serum HDL cholesterol levels ↑ Serum hs-CRP, plasma NO, and plasma TCA ↑	[68]
Type 2 diabetic patients with CHD	Probiotic and selenium	*L. acidophilus*, *L. reuteri*, *L.fermentum*, *B. bifidum*	Human	2 × 10^9^ CFU/g	12 weeks	Fasting plasma glucose, serum insulin levels, insulin resistance ↓ Triglycerides, VLDL and total cholesterol, and hs CRP ↓ NO ↑	[69]
Type 2 diabetic patients with CHD	Synbiotic	*L. acidophilus*, *L. casei*, *B. bifidum*	Human	2 × 10^9^ CFU/g	12 weeks	Fasting plasma glucose, serum insulin concentrations ↓ HLDL-cholesterol levels ↑	[70]
Overweight, diabetes, and CHD	Synbiotic	*L. acidophilus* strain T16 (IBRC-M10785), *L. casei* strain T2 (IBRC-M10783) *B. bifidum* strain T1 (IBRC-M10771)	Human	2 × 10^9^ CFU/g	12 weeks	Serum hs-CRP and plasma MDA ↓ Plasma NO ↑	[71]
Men with stable CAD	Probiotic	* L. plantarum * 299v (Lp299v)	Human	2× 10^10^ CFU/d	6 weeks	NO ↑ IL-8, IL-12, and leptin levels↓	[72]
CAD patients	Probiotic	* L. rhamnosus * GG (LGG)	Human	1.6 × 10^9^ CFU/d	12 weeks	IL1-Beta ↓ LPS ↓	[73]
CAD patients	Synbiotic	* L. rhamnosus * GG (LGG)	Human	1.9 × 10^9^ CFU/d	8 weeks	hs-CRP ↓ LPS ↓ TNF-α ↓	[74]
I schemic stroke	Probiotics	*C. butyricum*	Male ICR mice	No mention	2 weeks	MDA and TNF-α ↓ SOD activity ↑ Caspase-3, Bax and Bcl/Bax ↓ Butyrate contents ↑	[75]
Focal cerebral ischemia in mice	Probiotics	*B.**breve, L.**casei, L.**bulgaricus (L.**delbrueckii subsp. bulgaricus)*, *L. acidophilus*	Male BLC57 mice	10^7^ CFU/mL	14 days	Infarct size ↓ MDA and TNF-α ↓	[76]
Heat stroke in rats	Probiotics	* Bacillus licheniformis * strain (BL, CMCC 63516)	Male Sprague–Dawley (SD) rats	1 × 10^8^ CFU/mL	7 days	Hyperthermia, HS-induced death, multiple-organ injury, levels of serum inflammatory cytokines, and intestinal injury ↓ tight junctions ↑ Ratios of *Lactobacillus* and *Lactococcus* in gut microbiota ↑	[77]
Cerebral ischemia reperfusion injury in rats	Probiotics	*Lactobacillus* ILA amylovorus DSM 16698T (ILA)	Male SD rat	10^6^, 10^7^ and 10^8^ CFU/mL	2 h	Cerebral infarction volume and neural cell apoptosis ↓ MDA and TLR-4 ↓ SOD activity ↑ IkB and A20 ↑	[78]

hs-CRP, high sensitivity C-reactive protein; NO, nitric oxide; TAC, total antioxidant capacity; MDA, malondialdehyde; TLR-4, Toll-like receptor-4; SOD, superoxide dismutase; ↓, the levels, size or abundance decreased; ↑, the levels, size or abundance increased.

**Table 2 nutrients-13-02878-t002:** Beneficial effects of prebiotics on various CVDs.

Disease	Product	Prebiotics	Subject	Dose	Duration	Outcomes	Reference
Type 2 diabetic patients with CHD	Synbiotic	Inulin	Human	800 mg/day	12 weeks	Fasting plasma glucose, serum insulin concentrations ↓ HLDL cholesterol levels ↑	[70]
Overweight, diabetes, and CHD	Synbiotic	Inulin	Human	800 mg/day	12 weeks	Serum hs-CRP and plasma MDA ↓ Plasma NO ↑	[71]
CAD patient	Synbiotic	Inulin	Human	15 g/day	2 months	hs-CRP, LPS, TNF-α ↓	[74]
Chronic kidney disease patients	Prebiotic	Inulin	Human	19 g/day	6 months	Serum insulin and fasting glucose levels, HOMA-IR, total cholesterol, Triglycerides, CRP and homocysteine ↓ HDL cholesterol ↑	[122]
Women with type 2 diabetes	Prebiotic	Inulin and oligofructose	Human	10 g/day	8 weeks	Total antioxidant capacity ↑ Fasting plasma glucose, HbA1c, total cholesterol, LDL cholesterol, TC/HDL-c ratio, LDL-c/HDL-c ratio and malondialdehyde ↓	[123]
Hypercholesterolemic adults	Prebiotic	Concentrated oat beta-glucan with inulin and guar gum	Human	6 g/day	6 weeks	Total cholesterol and LDL cholesterol ↓ SCFAs ↑	[124]
Atherosclerosis in hypercholesterolemic APOE*3-Leiden mice	Prebiotic	Inulin	Male APOE*3-Leiden (E3L) mice	10% of diet	5 weeks	Atherosclerosis, number of macrophages, smooth muscle cells, collagen content, plasma total cholesterol levels ↑	[125]
Female rats with modeled heart failure	Prebiotics complex	Prebiotic complex based on fermented wheat bran	Female rat	0.5 g/day/rat	21 days	Concentrations of endotoxin, markers of *Lactobacilli*, and opportunistic microorganisms ↓ markers of *Bifidobacteria, Eubacteria*, and *Propionibacteria* ↑	[126]
Subjects with hypercholesterolemia	Prebiotic	Soluble fiber (minolest)	Human	16.5 g/day	3 months	Total cholesterol and LDL cholesterol ↓	[127]
A rat model of ischemia-reperfusion	Prebiotic	Polysaccharides of pectin (larch arabinogalactan)	Male SD rat	50 mg/kg/day	3 days	Gelsolin gene expression, p38 phosphorylation, apoptotic cells, and hif1-α gene expression ↓	[128]
CHD patients	Prebiotic	Chitosan oligosaccharides (COS)	Human	2 g/day	6 months	Blood urea nitrogen, serum creatinine, antioxidant levels, SOD and GSH ↑ ALT and AST ↓ Abundance of *Faecalibacterium, Alistipes*, and *Escherichia* ↑ Abundance of *Bacteroides, Megasphaera, Roseburia, Prevotella, and Bifidobacterium* ↓ Probiotic species *Lactobacillus*, *Lactococcus*, and *Phascolarctobacterium* ↑	[129]

HOMA-IR, homeostasis model assessment-estimated insulin resistance; HDL, high-density lipoprotein; LDL, low-density lipoprotein; TC/HDL-c ratio, total cholesterol to high-density lipoprotein cholesterol ratio; LDL-c/HDL-c ratio, low-density lipoprotein cholesterol to high-density lipoprotein cholesterol ratio; ↓, the levels, size or abundance decreased; ↑, the levels, size or abundance increased.

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
