# Peer review of "Potential Benefits of Probiotics and Prebiotics for Coronary Heart Disease and Stroke"

_nutrients, 2021, doi:10.3390/nu13082878_

Round 1

Reviewer 1 Report

The authors aimed to summarize current evidence regarding beneficial potentials of probiotics and prebiotics for prevention of CVD, with a focus on gut microbiota and immune system. Although this issue is timely and with clinical significance, this review is a good first draft that awaits the addition of some depth.  

Major points:

  1. I began reading this manuscript with enthusiasm. However, I was quite disappointed when I finished because there are several issues. First, the scope of the review is unclear. I thought it focuses on immune system in CVD, but as I read the manuscript, I feel like this is a review of inflammation and other mechanisms in CVD. For example, it also mentions things such HDAC, apoptosis, oxidative stress etc and their effects on CVDs. Please define clearly what the scope of the review is.
  2. The reference studies included in the review seem to be selective, but no clear selection criteria is provided, partly because the scope of the review is unclear. For example, the authors mentioned five studies were recruited from the database PubMed using keywords “probiotics and coronary heart disease” and listed in Table 1. Of note, references 74-78 are all coming from the same group (Raygan, F. et al). To my knowledge, there are other data in humans about probiotics and coronary heart disease (e.g. Moludi J et al., Nutr J. 2021 Jun 1;20(1):47. doi: 10.1186/s12937-021-00703-7; Moludi J, et al., Nutr Neurosci. 2021 Feb 28:1-10. doi: 10.1080/1028415X.2021.1889451; Hofeld BC, et al., Sci Rep. 2021 Feb 17;11(1):3972. doi: 10.1038/s41598-021-83252-7). As such, mentioning only one group seems to be inappropriate without strong rationale. In addition, there are more information from animal models and they need to be described in more details.
  3. Please conduct a more thorough literature review and include studies as relevant to the scope of this review. Also, please state all the strategy and timeframe of the searching. Please add a simple searching description- how many articles have been found, how and why papers are included/excluded, etc. It would be good to make a short comment on these information.
  4. Furthermore, CVDs are a cluster of disorders. Why the authors only discuss and summarize the role of probiotics/prebiotics on coronary heart disease and stroke? Hypertension is one of the most important risk factors for CVD, which is the leading cause of mortality. There are more data regarding hypertension, gut microbiota, and immune system. Unfortunately, none of these are mentioned in this review.
  5. There are several studies that are missing and I am not sure of the rationale for leaving them out. Whether maternal probiotics/prebiotics supplementations prevent CVDs in offspring is out of the scope of this review? If not, all related literature should be included and discussed.
  6. The authors' argument seems to be one-sided. They mentioned that supplementing probiotics/prebiotics is a good strategy for ameliorating CVDs. However, the more is not equivalent to the better as they might influence gut microbiota community. There are also negative studies have been published regarding the negligible effects associated with the use of probiotics/prebiotics as shown in CVDs. Please rephrase your argument with consideration of these questions.
  7. What does this review actually add to the research field of gut microbiota in CVD? A lot of the data mentioned in the manuscript are either already well-known or described in other recent review papers (Circ Res. 2017 Mar 31;120(7):1183-1196; J Am Coll Cardiol. 2019 Apr 30;73(16):2089-2105.; Oxid Med Cell Longev. 2020 Sep 26;2020:5394096; Nutrients. 2020 Feb 26;12(3):605; Food Funct. 2016 Feb;7(2):632-42, etc). The authors mentioned they mainly focused on immune system, but there seems no novel findings to provide an in-depth review. As such, I am not sure how this review on probiotics/prebiotics, gut microbiota and CVD in 2021 is different from other reviews.

Minor points:

Line 124: Firmicutes and Bacteroidetes should appear in Italic.

Line 192: CHD should be spelled out for the first time.

Table 1 and 2: For immune system and inflammation reaction, the strains and sex matters. Please provide the strains and gender of rats/mice in Table 1 and 2. Additionally, why synbiotic was repeatedly listed in Table 1 and 2? How to define the beneficial effect is from pre- or probiotics? Moreover, the outcomes listed in Table 1 and 2, they looks like mechanisms for me. Shouldn’t outcome mean cardiovascular outcome? Did probiotics/prebiotics really protect against CVDs?

Figure 1: Why SCFAs are not illustrated? Also, TMAO activates immune reaction is NOT the only way to induce CVDs.

Figure 2: Same question, why TMAO is NOT in figure 2? Prebiotics supplementation has been show to protect against CVDs.

I hope that the enclosed comments will be of help to the authors.

Author Response

We have studied the valuable comments and tried our best to revise the manuscript and address the comments. The point-to-point responses  are listed as below.

Comment 1: First, the scope of the review is unclear. I thought it focuses on immune system in CVD, but as I read the manuscript, I feel like this is a review of inflammation and other mechanisms in CVD. For example, it also mentions things such HDAC, apoptosis, oxidative stress etc and their effects on CVDs. Please define clearly what the scope of the review is.

Reply: Thank you for the comment. The scope of this review is to discuss the roles of probiotic and prebiotic in the development of CVD, with the focus on CHD and stroke. We mainly focus on two mechanisms, alterations of gut microbiota and immune responses (especially inflammatory responses).  However, other mechanisms, including anti-oxidation, HDAC, and apoptosis, were only briefly described.

Comment 2: The reference studies included in the review seem to be selective, but no clear selection criteria is provided, partly because the scope of the review is unclear. For example, the authors mentioned five studies were recruited from the database PubMed using keywords “probiotics and coronary heart disease” and listed in Table 1. Of note, references 74-78 are all coming from the same group (Raygan, F. et al). To my knowledge, there are other data in humans about probiotics and coronary heart disease (e.g. Moludi J et al., Nutr J. 2021 Jun 1;20(1):47. doi: 10.1186/s12937-021-00703-7; Moludi J, et al., Nutr Neurosci. 2021 Feb 28:1-10. doi: 10.1080/1028415X.2021.1889451; Hofeld BC, et al., Sci Rep. 2021 Feb 17;11(1):3972. doi: 10.1038/s41598-021-83252-7). As such, mentioning only one group seems to be inappropriate without strong rationale. In addition, there are more information from animal models and they need to be described in more details.

Reply: When we used keywords “probiotics and coronary heart disease” in PubMed, 64 related articles were found, within which only 5 of them mentioned the coronary heart disease directly in the title or abstract. 3 of them mentioned the disease coronary artery disease, which include the papers published by Moluji J’s group. Those 3 articles were missing in the previous submitted version of this review and have been added in lines 238-245, page 5 and Table 1 in this resubmitted version.

Comment 3: Please conduct a more thorough literature review and include studies as relevant to the scope of this review. Also, please state all the strategy and timeframe of the searching. Please add a simple searching description- how many articles have been found, how and why papers are included/excluded, etc. It would be good to make a short comment on these information.

Reply: Thank you for the comment. (1) All the studies relevant to the scope of this review have been added and updated in Tables 1 and 2 although the number of papers related to coronary heart disease and stroke is limited.  (2) All the strategies and timeframe of the searching have been stated in the parts “Beneficial effects of probiotics on CVD” in lines 225-228, page 5 and “Beneficial effects of prebiotics on CVD” in lines 170-172, page 13. We searched the relevant articles by typing keywords, selecting timeframe, and screening title and abstract.

Comment 4: Furthermore, CVDs are a cluster of disorders. Why the authors only discuss and summarize the role of probiotics/prebiotics on coronary heart disease and stroke? Hypertension is one of the most important risk factors for CVD, which is the leading cause of mortality. There are more data regarding hypertension, gut microbiota, and immune system. Unfortunately, none of these are mentioned in this review.

Reply: (1) We agree the comment that CVDs include many types of disorders involving the heart or blood vessels such as CHD, stroke, high blood pressure, venous thrombosis, thromboembolic disease, cardiomyopathy, and arrhythmia. The etiologies of different types of CVDs are also complicated. Due to the limitation of length and depth of discussion to be covered, we decided to focus on CHD and stroke as the representative CVD considering these two diseases are the most well-known and widely studied, yet the beneficial effects of probiotics and prebiotics on these two diseases were less addressed in the literature. (2) Hypertension is a very important risk factor of CVDs which has many published data. We considered that CHD and stroke are the most well-known and representative diseases of CVD, hence focused on these two diseases in this review. Nonetheless, we appreciated the clinical importance of hypertension on the progression to other more severe CVDs, hence the discussion about hypertension has been added in line 325-352, pages 20-21.  

Comment 5: There are several studies that are missing and I am not sure of the rationale for leaving them out. Whether maternal probiotics/prebiotics supplementations prevent CVDs in offspring is out of the scope of this review? If not, all related literature should be included and discussed.

Reply: Thank you for the comment. We have added several missing articles, which are related to the coronary artery disease in Table 1. The supplementations of maternal probiotics/prebiotics to prevent CVDs in the offspring are an interesting direction and important field which may be used to manage CVDs in young adults. In this review, however, we do not include this part into the scope so the relevant studies are not included.

Comment 6: The authors' argument seems to be one-sided. They mentioned that supplementing probiotics/prebiotics is a good strategy for ameliorating CVDs. However, the more is not equivalent to the better as they might influence gut microbiota community. There are also negative studies have been published regarding the negligible effects associated with the use of probiotics/prebiotics as shown in CVDs. Please rephrase your argument with consideration of these questions.

Reply: Thank you for the comment. The number of living microorganisms in foods containing probiotic at the time of human consumption should be above 106 cells/ml or cells/g according to WHO, and the therapeutic dose in clinical study is 108-109 cells/ml or cells/g in relevant reference. The doses of probiotics mentioned in this review do not exceed therapeutic dose (108-109). We agree that that consumption of more probiotics is not equivalent to better benefit human health, nor this statement is made throughout the review. One reference describing the controversy results was cited in this review, which demonstrated that inulin aggravated the accelerated atherosclerosis development driven by increased plasma cholesterol in the hypercholesterolemic APOE*3-Leiden mice (Ref 153). We have rephrased our conclusion in the part “Beneficial effects of prebiotics on CVD” in lines 196-197, page 14.

Comment 7: What does this review actually add to the research field of gut microbiota in CVD? A lot of the data mentioned in the manuscript are either already well-known or described in other recent review papers (Circ Res. 2017 Mar 31;120(7):1183-1196; J Am Coll Cardiol. 2019 Apr 30;73(16):2089-2105.; Oxid Med Cell Longev. 2020 Sep 26;2020:5394096; Nutrients. 2020 Feb 26;12(3):605; Food Funct. 2016 Feb;7(2):632-42, etc). The authors mentioned they mainly focused on immune system, but there seems no novel findings to provide an in-depth review. As such, I am not sure how this review on probiotics/prebiotics, gut microbiota and CVD in 2021 is different from other reviews.

Reply: Thank you for the comment. To add novelty to the research field of gut microbiota in CVD, (1) we have added more discussion about the metabolites produced by gut flora in lines 176-186, page 4, and lines 66-75, page 11. (2) Some metabolites produced by gut microbiota are neurotransmitters and play an important role in modulation of systemic inflammation, possibly through gut-brain axis. This part is discussed and added in lines 126-138, page 12, and lines 276-283, page 19. (3) We also discuss the role of postbiotics (gut microbiota-derived metabolites) in CVD in lines 296-324, page 19.

Minor points:

Line 124: Firmicutes and Bacteroidetes should appear in Italic.

Reply: The font have been revised to Italic.

Line 192: CHD should be spelled out for the first time.

Reply: This term has been spelled out in line 33, page 1.

Table 1 and 2: For immune system and inflammation reaction, the strains and sex matters. Please provide the strains and gender of rats/mice in Table 1 and 2. Additionally, why synbiotic was repeatedly listed in Table 1 and 2? How to define the beneficial effect is from pre- or probiotics? Moreover, the outcomes listed in Table 1 and 2, they looks like mechanisms for me. Shouldn’t outcome mean cardiovascular outcome? Did probiotics/prebiotics really protect against CVDs?

Reply: Thank you for the comments. (1) The strains and sex of rats/mice have been added in Table 1 and 2. (2) Since synbiotic is a combination of probiotics and prebiotics, we listed synbiotic in both tables with the detailed components of symbiotic listed in the two tables individually. (3) Considering the total- and low-density lipoprotein (LDL)-cholesterol, high sensitivity C-reactive protein (hs-CRP) are the main cardiovascular markers, hence we use these biomarkers to represent the results of CVDs. (4) Most of the literature support the idea that the prebiotics/probiotics could prevent CVDs, yet controversial results are available. Considering many of the studies were done on different animal models, human subjects with diversified background such as age, geography, diet habit, genetics, and other factors, we do not intend to promote the idea that prebiotics/probiotics is the “cure” or completely prevent CVDs to occur. Instead, supplementation of prebiotics/probiotics could be used as one of the strategies to manage CVDs.

Figure 1: Why SCFAs are not illustrated? Also, TMAO activates immune reaction is NOT the only way to induce CVDs.

Reply: Thank you for the comment. The metabolites SCFAs and secondary bile acids were added in Figure 1.

Figure 2: Same question, why TMAO is NOT in figure 2? Prebiotics supplementation has been show to protect against CVDs.

Reply: In the discussion of prebiotics, fiber is one of the most popular prebiotics and SCFAs are the end-product of fiber produced by gut microbiota. We totally agree that TMAO is an important metabolite associated with the progression of CVDs. Thus, we added this metabolite in Figure 2.

Reviewer 2 Report

The manuscript by Wu and Chiou brings an interesting discussion on the potential use of microbiota-based interventions in the treatment of cardiovascular disorders. The topic is pertinent to the general audience of this journal. However, this reviewer has some further recommendations:

In general, the content of the review is good. The authors seem to believe that the effects of the gut microbiome to CVDs are mediated by two main mechanisms 1 ) immune system 2) metabolism. I would recommend adding a more generic figure that illustrates that.

The sessions would also benefit from organization :

  • The immune system seems to be key, and this part should be expanded to include more specific sof probiotics and prebiotics to cardiovascular disease that are mediated by the immune system.
  • As for the metabolism I recommend adding more references from metabolome and metagenome studies in CVDs. Further, It would be also interesting to discuss if there is any effect from the gut microbiome on modulating lipid metabolism which is important for CVDs.

Additional comments

Add definition for probiotics and prebiotics earlier in the text . Perhaps line 37 rather than 159

Sentence line 40 and 41 : “Besides the intestinal tract …” is very confusing please rephrase it

Lines 45 to 47 . It would be important to add original papers that support the effects of the gut microbiome on” lowering cholesterol levels, attenuating oxidative stress etc …”.

I recommend replacing gut flora for gut microbiota and being consist throughout

Lines 48 – 49 – have been widely studied replace it for are being widely studied

For the general public , what can be considered a functional food ?

Akkermansia muciphila cannot be considered a next-generation probiotics. Instead, A.  muciphila is also associated with some other diseases such as MS, PD etc….As it is context dependent ,  I would tone down this statement. The same should be applied for “protective metabolite butyrate” and “harmful metabolite TMAO”. Everything seems to be context dependent.

Lines 192 – 193 , “ five studies , searched ..” the methodology can be omitted here as it interrupts the flow of the narrative.

SOD- spell out

Author Response

We have studied the valuable comments and tried our best to revise the manuscript and address the comments. The point-to-point responses are listed as below.

Major concern from reviewer#2

Comment 1: In general, the content of the review is good. The authors seem to believe that the effects of the gut microbiome to CVDs are mediated by two main mechanisms 1) immune system 2) metabolism. I would recommend adding a more generic figure that illustrates that.

Reply: Thanks for your valuable comment. A generic figure has been added as Figure3 in page 22.

Comment 2: The immune system seems to be key, and this part should be expanded to include more specific probiotics and prebiotics to cardiovascular disease that are mediated by the immune system.

Reply: Thank you for the comment. The content about beneficial effects of specific probiotics and prebiotics in cardiovascular disease from the view of immune system has been added in lines 116-121, pages 12 and lines 290-292, page 20.

Comment 3: As for the metabolism I recommend adding more references from metabolome and metagenome studies in CVDs. Further, it would be also interesting to discuss if there is any effect from the gut microbiome on modulating lipid metabolism which is important for CVDs.

Reply: Thank you for the valuable comment. (1) Metabolome and metagenome studies in CVDs are interesting. We added more discussions about the metabolites produced by gut microbiota in 176-186, page 4, lines 66-75, page 11. (2) As suggested by reviewer#1 and reviewer#3, we added the effects of probiotic and prebiotic in hypertension in lines 325-352, pages 21. The potential effect of gut microbiota on the lipid metabolism was discussed on the role of probiotics on converting the cholesterol into secondary bile salts which projects their potential effect on the lipid metabolism as stated in lines 182-186, page 4, lines 215-223, page 5, and lines 66-70, page 11.   

Minor points:

  1. Add definition for probiotics and prebiotics earlier in the text. Perhaps line 37 rather than 159

Reply: The definition of probiotics and prebiotics have been added to the introduction.

  1. Sentence line 40 and 41: “Besides the intestinal tract …” is very confusing please rephrase it

Reply: We rephrase this sentence into “Besides the intestinal dysfunctions, accumulating evidence suggests that both of probiotic and prebiotic could ameliorate metabolic disorders including obesity, diabetes, and CVD” in line 56, page 2.

  1. Lines 45 to 47. It would be important to add original papers that support the effects of the gut microbiome on” lowering cholesterol levels, attenuating oxidative stress etc …”.

Reply: 2 more references have been added in this part in line 64, page 2.

  1. I recommend replacing gut flora for gut microbiota and being consist throughout.

Reply: Thank you for the comment. The term has been revised as advised.

  1. Lines 48 – 49 – have been widely studied replace it for are being widely studied

Reply: Thank you for the comment. This sentence has been revised as advices.

  1. For the general public, what can be considered a functional food ?

Reply: The definition of functional food is “foods or dietary components that may provide a health benefit beyond basic nutrition” by International Food Information

Council (IFIC). The functional foods exhibiting potential cardiovascular protective effects include nuts (bioactive compounds: tocopherols, omega-3 fatty acids), legumes (bioactive compounds: fiber and polyphenols), fish oil (bioactive compound: omega-3 fatty acids), fruits and vegetables (bioactive compound: vitamin C), dark chocolate (bioactive compound: flavonoid), and grapes (bioactive compounds: anthocyanins, catechins, cyanidins, and flavonols).

  1. Akkermansia muciphila cannot be considered a next-generation probiotics. Instead, A. muciphila is also associated with some other diseases such as MS, PD etc….As it is context dependent, I would tone down this statement. The same should be applied for “protective metabolite butyrate” and “harmful metabolite TMAO”. Everything seems to be context dependent.

Reply: Thank you for the comment. Zhai Q,et al. (Crit Rev Food Sci Nutr. 2019;59(19): 3227-3236. doi: 10.1080/10408398.2018.1517725.) first used the term “next-generation probiotic” to describe Akkermansia municiphila. Yet indeed this concept is not scientifically accepted and included in the probiotic list yet, according to WHO. We have rephrased the sentence in lines 148-150, page 3. Similarly, the term of “protective metabolite butyrate” and “harmful metabolite TMAO” have been revised as suggested in lines 169-170, page 4.

  1. Lines 192 – 193, “ five studies , searched ..” the methodology can be omitted here as it interrupts the flow of the narrative.

Reply: As Reviewer#1 suggested us to add the detailed methodology of searching, we still keep the methodology and

  1. SOD- spell out

Reply: SOD has been spelled out when this term first appeared in this review in line 258-259, page 6.

Reviewer 3 Report

In this article, the authors review the published literature on the potential beneficial effects of prebiotics and probiotics on cardiovascular disease. The researchers focus primarily on coronary heart disease and stroke, and among the possible mechanisms by which these nutritional supplements may exert their effects, the authors highlight the changes they may cause in the microbiota and inflammatory response.

This is a well-written study: precise, clear, and presented in a simple way. The authors discuss the main mechanisms that have been proposed to exert the beneficial effects of probiotics and prebiotics. However, all of them are explained superficially and in little depth. While the study is intriguing, there are other several concerns:

Major comments:

- Although the authors use a large number of references in the review, the number of articles analyzing prebiotics, probiotics, and their relation to cardiovascular diseases are rather scarce.

- In the first part of the article, it is interesting how the authors focus on the effects observed in two cardiovascular diseases such as coronary heart disease and stroke. However, in the second part, the researchers discuss numerous pathologies without focusing on any of them. In this case, the existing literature is more extensive than the authors summarise, and it is not clear on what basis they have chosen these articles and not others. Furthermore, this means that there is no homogeneity in the review and the data are very scattered in the second part of the article. Perhaps it would be interesting if the authors focused only on the effects on both diseases throughout the review and if the title did not name all cardiovascular diseases but only these two.

- It would be interesting if the authors, in addition to discussing changes in the microbiota and inflammatory response, would discuss the antioxidant effects of probiotics as a possible explanation for the beneficial effects observed at the cardiovascular level.

- Since the authors explain the participation of metabolites produced by bacteria such as TMA and short-chain fatty acids, they could also summarize the participation of lipopolysaccharide (LPS) and its translocation to the systemic circulation.

- Another important topic that are not explained in this review would be the gut microbiota-brain axis and the interaction gut microbiota-sympathetic nervous system.

- Although justified by the authors in lines 217-219, summarising the effects of probiotics and prebiotics on hypertension and endothelial dysfunction is necessary for a review of this kind.

Minor comments:

- Line 213 (page5): the font size of Lactobacillus is bigger.

- In line 28 (on page 9) the authors explain that the phylum Bacteroidetes was increased in sick animals and humans. However, they state on line 40 that the administered probiotics caused the increase of Bacteroidetes as well. Is this correct? And, how do the authors explain the change in the Firmicutes/Bacteroidetes ratio?

- Line 32-34 (page 9): The authors indicate that nutraceuticals evoke more interest than pharmaceuticals agents. On what basis do they make this claim?

- Line 49 (page 9): Authors should indicate what the probiotic mixture VSL#3 consists of.

The authors have written an interesting and well-structured review, but I consider that it does not bring anything new among all the very similar reviews that have been published in recent years.

Author Response

We have studied the valuable comments and tried our best to revise the manuscript and address the comments. The point-to-point responses are listed as below.

Comment 1: Although the authors use a large number of references in the review, the number of articles analyzing prebiotics, probiotics, and their relation to cardiovascular diseases are rather scarce.

Reply: Thanks for the comment. We have revised the title to better clarify the scope of this review. In this review, we focus on two well-known CVDs, CHD and stroke, the total number of published papers analyzing probiotics, prebiotics, and their relation to these two diseases in PubMed is only 23. We listed all the article in the tables and references list.

Comment 2: In the first part of the article, it is interesting how the authors focus on the effects observed in two cardiovascular diseases such as coronary heart disease and stroke. However, in the second part, the researchers discuss numerous pathologies without focusing on any of them. In this case, the existing literature is more extensive than the authors summarise, and it is not clear on what basis they have chosen these articles and not others. Furthermore, this means that there is no homogeneity in the review and the data are very scattered in the second part of the article. Perhaps it would be interesting if the authors focused only on the effects on both diseases throughout the review and if the title did not name all cardiovascular diseases but only these two.

Reply: As CVD is a group of disorders, and a lot of risk factors could contribute to CVD, such as hypertension, smoking, diabetes mellitus, obesity, unhealthy diet, cholesterol and lipids, depression and anxiety, etc, we could not discuss all CVDs and risk factors in this review. Thus, we focus on two CVD, CHD and stroke. To let people know more about CVD at the first, therefore, we also discussed other CVDs except CHD and stroke. In the following paragraphs when discussing probiotics and prebiotics, we illustrate how to choose articles relevant to the scope of our review, which is shown in lines 225-228, page 5 and in lines 170-172, page 13.  

Comment 3: It would be interesting if the authors, in addition to discussing changes in the microbiota and inflammatory response, would discuss the antioxidant effects of probiotics as a possible explanation for the beneficial effects observed at the cardiovascular level.

Reply: Thank you for the valuable comments. We agree that the antioxidant effects of probiotics in CVD are very important. One review (Vasquez, EC. et al, 2019) has discussed that probiotics as beneficial dietary supplements could prevent and treat cardiovascular disease from the view of uncovering their impact on oxidative stress. The theme of our review which is invited is to discuss role of probiotics and prebiotics in CVD, with the focus on CHD and stroke, via gut microbiota and immune modulation.

Comment 4: Since the authors explain the participation of metabolites produced by bacteria such as TMA and short-chain fatty acids, they could also summarize the participation of lipopolysaccharide (LPS) and its translocation to the systemic circulation.

Reply: Thank you for the comment. LPS is a cell wall component of Gram-negative bacteria (e.g. E. coli), hence we do not include it as a metabolite produced by gut bacteria. Yet indeed it is true that microbially-derived LPS may compromise cardiovascular function and increase CVD risk. The discussion was added into the part “Gut microbiota and CVD” in lines 176-182, page 4.

Comment 5: Another important topic that is not explained in this review would be the gut microbiota-brain axis and the interaction gut microbiota-sympathetic nervous system.

Reply: Thank you for the valuable comment. The parts discussing gut microbiota-brain axis have been added in lines 126-138, page 12 and lines 276-283, page 19.

Comment 6: Although justified by the authors in lines 217-219, summarising the effects of probiotics and prebiotics on hypertension and endothelial dysfunction is necessary for a review of this kind

Reply: Thank you for the comment. The discussion about the effects of probiotics and prebiotics on hypertension has been added in lines 325-352, page 20-21.

Minor comments:

Line 213 (page 5): the font size of Lactobacillus is bigger.

Reply: The front size has been revised.

In line 28 (on page 9) the authors explain that the phylum Bacteroidetes was increased in sick animals and humans. However, they state on line 40 that the administered probiotics caused the increase of Bacteroidetes as well. Is this correct? And, how do the authors explain the change in the Firmicutes/Bacteroidetes ratio?

Reply: Thank you for the comment. The Firmicutes/Bacteroidetes (F/B) ratio is widely accepted to have an important influence in maintaining normal intestinal homeostasis. Increased or decreased F/B ratio is regarded as dysbiosis. The former is usually observed with obesity, and the latter with inflammatory bowel disease (IBD). CVD is a complex disorder and hence we could not conclude which phylum is abundance in this type of disease. The chances are that F/B ratio may change in different CVDs. Thus, the results of Bacteroidetes’ abundance might not be the same in different types of CVD discussed in our review.

Line 32-34 (page 9): The authors indicate that nutraceuticals evoke more interest than pharmaceuticals agents. On what basis do they make this claim?

Reply: There are several publications have discussed the nutraceuticals as therapeutic agents for atherosclerosis (J. W. E. Moss, D. P. Ramji, Nat. Rev. Cardiol. 2016, 13, 513. doi: 10.1038/nrcardio.2016.103; J. W. E. Moss, J. O. Williams, D. P. Ramji, Biochim. Biophys. Acta. 2018, 1864, 1562.doi: 10.1016/j.bbadis.2018.02.006; VL O'Morain, DP Ramji, Mol Nutr Food Res, 2020, 64(4):e1900797. doi: 10.1002/mnfr.201900797). All of them have indicated that nutraceuticals evoke more interest than pharmaceuticals agents, hence we adopt the concept into this review.

Line 49 (page 9): Authors should indicate what the probiotic mixture VSL#3 consists of.

Reply: Thank you for the comment. The content of VSL#3, which is a mixture of S. thermophilus, B. breve, B. bacterium longum, B. infantis, L. acidophilus, L. plantarum, L. paracasei and L. delbrueckii subsp. bulgarius, has been added in this review in line 50-51 page 10.

Round 2

Reviewer 1 Report

The authors have made adequate changes to the manuscript and figures. The second edition of the paper is well revised and I believe it is acceptable for publication.

Author Response

Thank you for the valuable comments. We appreciate this opportunity to have this review published in your esteemed journal.